# Efficient Learning of Sparse and Decomposable PDEs using Random Projection

**Md Nasim**[1]          **Xinghang Zhang**[2]          **Anter El-Azab**[2]          **Yexiang Xue**[1]

[1]Department of Computer Science, Purdue University, West Lafayette, IN, USA
[2]School of Materials Engineering, Purdue University, West Lafayette, IN, USA

## Abstract

Learning physics models in the form of Partial Differential Equations (PDEs) is carried out through back-propagation to match the simulations of the physics model with experimental observations. Nevertheless, such matching involves computation over billions of elements, presenting a significant computational overhead. We notice many PDEs in real world problems are sparse and decomposable, where the temporal updates and the spatial features are sparsely concentrated on small interface regions. We propose RAPID-PDE, an algorithm to expedite the learning of sparse and decomposable PDEs. Our RAPID-PDE first uses random projection to compress the high dimensional sparse updates and features into low dimensional representations and then use these compressed signals during learning. Crucially, such a conversion is only carried out once prior to learning and the entire learning process is conducted in the compressed space. Theoretically, we derive a constant factor approximation between the projected loss function and the original one with poly-logarithmic number of projected dimensions. Empirically, we demonstrate RAPID-PDE with data compressed to 0.05% of its original size learns similar models, compared with uncompressed algorithms in learning a set of phase-field models, which govern the spatial-temporal dynamics of nano-scale structures in metallic materials.

## 1 INTRODUCTION

Learning physics models in the form of Partial Differential Equations (PDEs) have numerous applications in the field of physics, engineering and life sciences. Examples include the heat and wave equations, Schrodinger's equation, Navier-Stokes equation, etc. These PDEs are essential descriptors of many complex dynamic processes and physical phenomena. Identifying PDEs automatically from experiment data has attracted great interest recently in the field of AI driven scientific discovery (Xue et al. [2021], Long et al. [2018], Sirignano and Spiliopoulos [2018], Qian et al. [2020], Lagergren et al. [2020]). PDEs can be learned via minimizing the mismatch between the ground-truth dynamics reflected in the experiment data and the predicted dynamics, which is typically simulated using a neural network with the current PDE parameters (Xue et al. [2021], Long et al. [2018]). Back-propagation is then used to minimize such differences by updating the PDE parameters. Despite its empirical success, back-propagating the loss function typically involves operations over billions of mutually interacting elements, hence yielding a heavy computational overhead.

We propose a novel randomized algorithm to speed up the learning of PDEs from experiment data. In particular, we notice the *sparse and decomposable* nature of many real-world PDE systems. More precisely, we find the temporal updates of many PDE systems can be *decomposed* into parameter functions of several *sparse features*. This is not a coincidence - the sparsity and decomposability of PDEs are natural consequences of the interface problems, where PDE solution updates are concentrated in small interface regions, where components of different physical properties meet each other. Outside of the interface regions, the updates are almost all zero, resulting in rather sparse updates.

We propose Random Projection Based Efficient Learning of Sparse and Decomposable PDEs (RAPID-PDE), exploiting the sparse nature of the learning problems. Our method is inspired from the idea of compressed sensing (Donoho [2006], Candes and Tao [2006], Candès et al. [2006]), relying on the intuition that as the system changes are sparse, therefore the change in high dimensional state vectors can be represented in an efficient manner by projecting into a lower dimensional space. Different from compressed sensing, which consists of the compression and recovery phases, RAPID-PDE compresses the updates and features into a

*Accepted for the 38th Conference on Uncertainty in Artificial Intelligence* (UAI 2022).

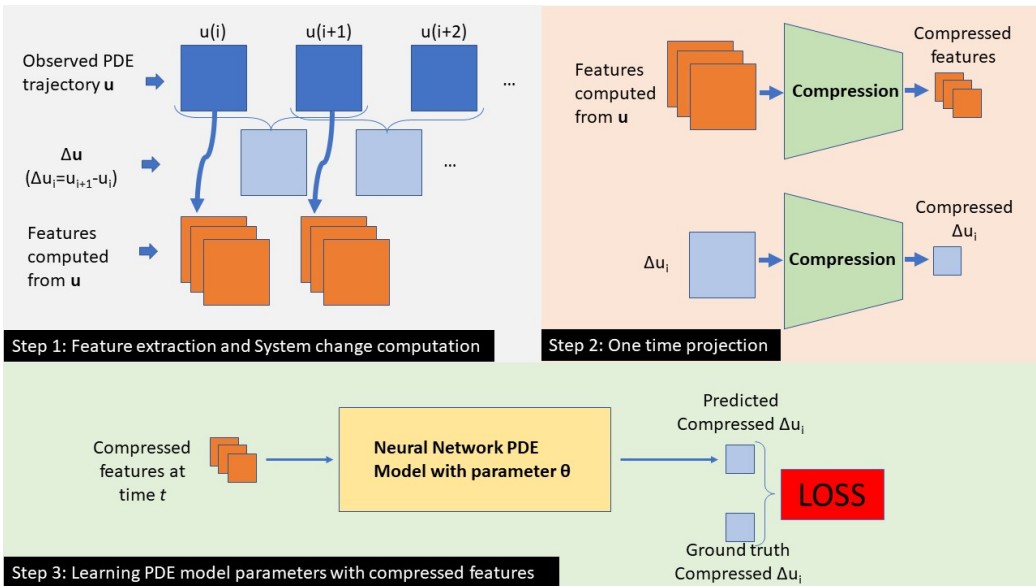

Figure 1: High-level idea of RAPID-PDE. Step 1: from the extracted PDE trajectories from data, we compute a number of features and also the difference between consecutive PDE trajectories. Step 2: we use random projection to project the high dimensional features and trajectory changes to low dimensional space. Step 3: Backpropagation is used to minimize the difference between the predicted and the ground-truth changes in the compressed space.

reduced space *once as a pre-processing step* before learning. The entire learning process is then carried out in the compressed space with no need for converting back to the original space.

The high-level idea of RAPID-PDE is shown in Figure 1. From the PDE trajectories obtained from observations, we first extract a number of sparse features and also the updates from experiment data. In the second step, we compress both the features and the updates using random projection into low dimensional space. The third step is the final learning step, where back-propagation is used to minimize the difference between the predicted updates and the ground-truth updates *in the compressed space*.

Theoretically, we show the loss function optimized by RAPID-PDE is at most $(1 + \delta)^2$, *a constant approximation factor* times the original loss function, with probabilities scales in the order of $1 - \Omega((1/\delta)^{2k} \exp(-n\delta^2))$. Here, $n$ is the projected dimensionality and $k$ is the number of non-zero elements in the updates. This ensures $O(k/\delta^2 \log(1/\delta))$ of projected dimensions are sufficient for $(1 + \delta)^2$ approximations.

Our RAPID-PDE algorithm is used to speed up the learning of nano-structure evolution in engineering materials in action. In particular, we focus on learning the physics rules which govern the dynamics of void defect evolution (Figure 2I) and the grain growth (Figure 2II). Experimental results show that, compared to a baseline method with no compression, RAPID-PDE can reduce the training times for learning the phase-field models while preserving the qual-

ity of learned models. RAPID-PDE can reduce the training times by as much as $70\%$ for grain growth model and by nearly $50\%$ for void evolution dynamics, when the data is compressed to $0.05\%$ of the original size. Testing with a separate test set, we find the mean squared error (MSE) for both applications after 100 steps of simulation to be very small, suggesting little to no loss in learning performance after compression. Upon simulation, we find that the simulated output with trained model parameters matches closely with the outputs simulated with ground truth physics model.

Our contribution is as follows: 1) We introduce sparse and decomposable PDEs as a special class of PDEs and show the sparse and decomposable nature of many real-world PDEs, 2) we propose RAPID-PDE, an efficient method based on random projection to learn model parameters of sparse and decomposable PDEs, 3) we provide a theoretic analysis on the effect of compression on the loss function used to update model parameter, 4) we show the efficacy of our proposed solution for two cases - learning the PDE-based phase-field model parameters for void defect evolution and grain growth in materials science.

## 2 BACKGROUND

### 2.1 DYNAMIC SYSTEMS REPRESENTED IN PARTIAL DIFFERENTIAL EQUATIONS

Partial Differential Equations (PDEs) are mathematical equations involving partial derivatives of multi-variable functions. Multi-variable PDEs are ubiquitous in physics

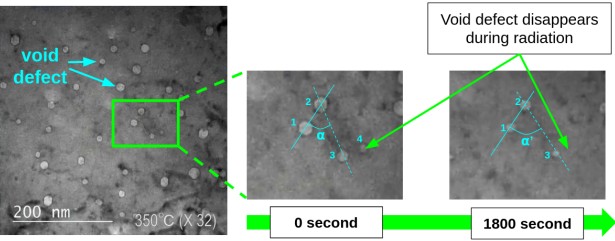

(I) Nanovoid Defects in Crystalline Materials

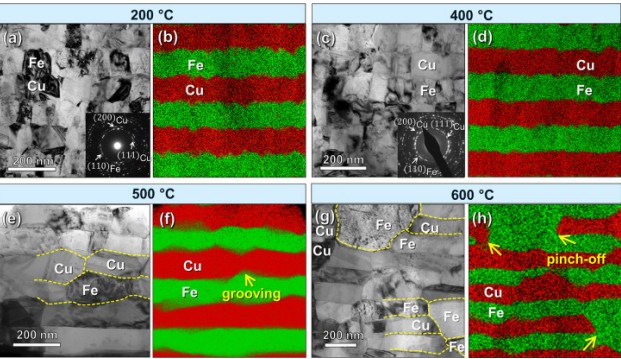

(II) Grain growth in Materials during Annealing (Niu et al. [2021])

Figure 2: Void defect evolution and grain growth in engineering materials are real world phenomena learnable by RAPID-PDE. (I) Void shaped defects in a Cu specimen at $350°$. These defects are dynamic and evolve as such size and position change, as shown by change of $\alpha$ to $\alpha'$ between voids 1,2 and 3, and disappearance of void 4. (II) Grain growth in Cu/Fe 100 nm multilayer upon annealing at different temperatures. XTEM micrographs and EDS maps show the microstructure evolution at temperatures 200°C (a-b), 400°C (c-d), 500°C (e-f), 600°C (g-h). Grooving and grain growth are observed at 500°C, while layer pinch-off occurred after 600°C annealing.

and engineering disciplines, and usually involve time and space. The order of a PDE is the order of the highest partial derivative term in the equation. In practice, we mostly encounter first or second order differential equations. In general, a second order PDE has the following general form:

$$\frac{\partial u(\vec{p}, t)}{\partial t} = \sum_i M_i(u)\nabla^2 F_i(u) + \sum_j N_j(u)\nabla G_j(u) + D(u)$$

(1)

Here, $u(\vec{p}, t)$ represents state of the system and is a function of both spatial coordinate $\vec{p} \in \mathbb{R}^d$ and time $t$. $\nabla$ represents the first order spatial derivative, while $\nabla^2 = \nabla.\nabla$ represents the second order spatial derivative Laplace operator. The functions $M, N, F, N, G, D$ can be either linear or non-linear in $u$.

## 2.2 PHASE-FIELD MODELING

We focus on phase-field models as motivating examples in this paper, where the state of a physical system is described using a set of phase-field variables. We point out that our computational approaches based on random projection generalize to other PDE systems as well. In phase-field models, the phase-field variables vary rapidly along phase boundaries. In this paper, we consider two example phase-field modeling for real world application - **grain growth** and **nanovoid defect evolution** in crystalline materials.

### 2.2.1 Case Study 1: Grain Growth

Polycrystalline materials used in many engineering applications often contain multiple grains with different crystal orientations. These grains evolve over time, some grow bigger and some shrink, thus changing their interface boundaries. The understanding of the grain growth dynamics is of high interest to physicists, as the dynamics affect the physical and mechanical properties of the material.

Many models have been proposed to model this grain growth phenomenon. For this study, we focus on the phase-field model proposed in Fan and Chen [1997]. In this phase-field model of grain growth, each grain $i$ is represented by $\eta_i$, which takes the value of 1 inside $i$-th grain, 0 outside the grain and lies in $[0, 1]$ at the grain boundary. The evolution of $\eta_i$ is described by the Allen-Cahn equation for non-conserved variables:

$$\frac{\partial \eta_i}{\partial t} = L\frac{\partial F}{\partial \eta_i}, \quad i = 1, 2, 3, \dots, N$$

(2)

Here, $L$ is the mobility coefficient and $F$ is the free energy. Writing out the expression for $F$ and taking derivatives, this governing equation can be written as follows:

$$\frac{\partial \eta_i}{\partial t} = L(1 - \eta_i{}^2 - 2\sum_{i\neq j}\eta_j{}^2)\eta_i + L\kappa\nabla^2\eta_i.$$

(3)

Here, $\kappa$ represents the gradient coefficient. These scalar parameters $L, \kappa$ affect the way grain volume and shape change over time and are the parameters to be learned in this paper. Fitting into the general form given in Equation 1, $\eta_i$ is the variable $u$. $D(.)$ takes the form $L(1 - \eta_i{}^2 - 2\sum_{i\neq j}\eta_j{}^2)\eta_i$. $M_1(.)$ is $L\kappa$ and $F_1(.)$ is $\eta_i$.

### 2.2.2 Case Study 2: Nanvoid Defect Evolution

Nano-sized void defects and dislocation loops are common phenomena in materials under extreme condition - high heat and irradiation. Over time, these void defects cause significant microstructure evolution and consequently degradation of materials. The evolution of these void defects is of high interest to physicists, for designing materials that can better withstand the extreme environments.

In this paper, we focus on a simplified version of the prominent and widely used void defect evolution model as described in Millett et al. [2011]. In this model, the state of the material is represented by three phase-field variables $c_v, c_i$ and $\eta$. These phase-field variables are continuous, lie in range $[0, 1]$ and vary rapidly at the interfaces. $c_v$ represents the percentage of void defects in crystal lattice, resulting from the absence of atoms at certain crystal lattice locations. $c_v$ takes the value of 1 inside void regions, 0 outside void regions and values in range $[0, 1]$ at the interface of void regions. Similarly, $c_i$ represents the percentage of interstitial, another type of crystal defect that results from the presence of atoms in a normally unoccupied location in the lattice. $c_i$ is 1 inside interstitial region, 0 outside and $[0, 1]$ at the interface boundary. The phase-field variable $\eta$ is also continuous, differentiating the two phases - solid ($\eta = 0$) and void ($\eta = 1$). The evolution of the phase-field variables minimizes the total free energy $F$:

$$F = N \int_V \left[ h(\eta)f^s(c_v, c_i) + j(\eta)f^v(c_v, c_i) + \frac{\kappa_v}{2}|\nabla c_v| + \frac{\kappa_i}{2}|\nabla c_i| + \frac{\kappa_\eta}{2}|\nabla \eta| \right] \mathrm{d}V.$$

The standard equation for temporal updates to $c_v, c_i$ and $\eta$ are as follows:

$$\frac{\partial c_v}{\partial t} = \nabla \cdot (M_v \nabla \frac{1}{N} \frac{\delta F}{\delta c_v}),$$
$$\frac{\partial c_i}{\partial t} = \nabla \cdot (M_i \nabla \frac{1}{N} \frac{\delta F}{\delta c_i}),$$
$$\frac{\partial \eta}{\partial t} = -L \frac{\delta F}{\delta \eta}.$$

Here, $M_v, M_i$ are diffusivities of voids and interstitials dependent upon material property, $L$ is the mobility coefficient. $\frac{\partial F}{\partial c_v}, \frac{\partial F}{\partial c_i}, \frac{\partial F}{\partial \eta}$ are functional derivatives of total free energy $F$ of the system. All of these above expressions for the temporal updates of the phase-field variables can be rearranged and written in the form as Equation 1. We leave such derivations to the supplementary materials. The scalar parameters $M_v, M_i, L, \kappa_v, \kappa_i, \kappa_\eta$ and parameters included in $f^s, f^v$ control system dynamics, and these are the parameters we learn from experiment data.

### 2.2.3 Finite Difference for PDE Simulation

Finite difference method is a widely used technique to simulate PDEs in discrete form. Using this technique, we divide space into meshes, and replace the derivatives with finite difference quotients. For example, the derivative with respect to time $\frac{\partial u(\vec{p}, t)}{\partial t}$ can be approximated using $\frac{u(\vec{p}, t+d_t) - u(\vec{p}, t)}{d_t}$, where $d_t$ is a small constant. The spatial derivative can be approximated similarly. For example, $\frac{\partial u}{\partial x}(x, y, t)$ can be approximated by $\frac{u(x+d_s, y, t) - u(x, y, t)}{d_s}$.

## 3 SPARSE DECOMPOSABLE PDES

Our RAPID-PDE algorithm is based on an interesting observation that the governing equations of many real-world PDE systems can be written in the following *sparse and decomposable* form, i.e., as a linear combination of sparse features times coefficient terms:

$$\frac{\partial u(\vec{p}, t)}{\partial t} = [\phi_1(\theta), \phi_2(\theta), \dots, \phi_n(\theta)] \begin{bmatrix} W_1(u) \\ W_2(u) \\ \vdots \\ W_n(u) \end{bmatrix} \quad (4)$$

We would like to point out two aspects of this formulation: (i) *separation of learning parameters from observational features*: in this form, $\theta$ are the parameters to be learned. Only $\phi_1(\theta), \dots, \phi_n(\theta)$ depend on $\theta$, while $W_1, \dots, W_n$, only depends on the current system state $u$. (ii) *sparse nature of observational features*: We observe that observational features $W_1, \dots, W_n$ are sparse in nature, with very few non-zero entries. This is actually not a coincidence. Rather, it is because of the nature of our interface problems, where most of the updates are around the interfaces of different material compositions. Such interfaces, by nature, only account for a small region of the entire space.

**Sparse Decomposition for Grain Growth**. For example, in the phase-field model of grain growth in materials with $d$-grains, the temporal evolution of the phase-field variable describing the $i$-th grain $\eta_i$ is described by Equation 3:

$$\frac{\partial \eta_i}{\partial t} = L(1 - \eta_i{}^2 - 2\sum_{i \neq j}\eta_j{}^2)\eta_i + L\kappa \nabla^2 \eta_i$$

Comparing this with Equation 4, we can write the following:

$$\phi_{i,1} = L, \quad \phi_{i,2} = L\kappa,$$
$$W_{i,1} = (1 - \eta_i{}^2 - 2\sum_{i \neq j}\eta_j{}^2)\eta_i, \quad W_{i,2} = \nabla^2 \eta_i.$$

We use subscript $i$ to indicate this equation holds for evolution of the $i$-th grain. Inside grain $i$, only $\eta_i$ is 1, the rest are 0. Hence, $W_{i,1}$ is zero because the terms inside the summation is just the sum of 0s, while the $\eta_i^2$ cancels out the 1 at the beginning. The multiplication with $\eta_i$ ensures that $W_{i,1}$ is 0 outside the $i$-th grain as well. The other feature $W_{i,2}$ is 0 both inside and outside grain $i$, because second order Laplacian is 0 when applied to a region of the same value. We can see a visual representation of these $W_{i,1}, W_{i,2}$ features for grain growth in Figure 3I. In summary, we can see that the computed features are all sparse and only non-zero at the interface boundaries of the grains.

We leave the sparse decomposition of the nanovoid evolution to the supplementary materials. Despite a more complex form, the PDEs describing nanovoid evolution can be

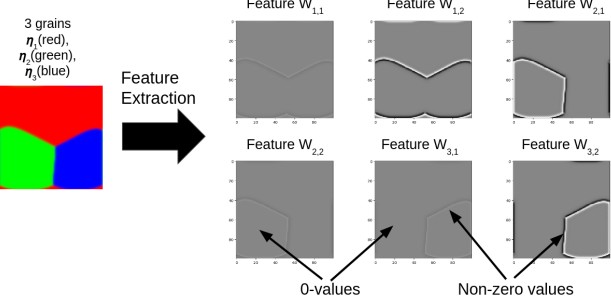

(I) Feature Extraction

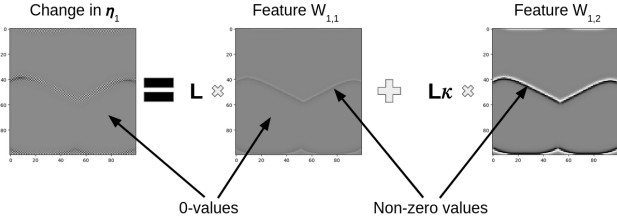

(II) Relation between system state change and features

Figure 3: The PDEs for grain growth can be decomposed into a couple of sparse features. We show such feature computation for grain growth in materials with 3 grains, represented by 3 different colors. (I) Sparse features computed from the system state variable $\eta$, (II) Change in $\eta_1$ represented by sum of sparse features.

written in the sparse and decomposable form as well. All observational features are also sparse.

# 4 LEARNING SPARSE DECOMPOSABLE PDE WITH RANDOM PROJECTION

## 4.1 LEARNING VIA MATCHING EXPERIMENTS AND SIMULATION

Data-driven scientific discovery attracted recent interest to learn physics models automatically from data. The learning of physics systems can be achieved in the matching of forward simulations of the current physics model with data from physical experiments. In particular, the data collected from physics experiments are typically in the form of $\{u_{t_1}, u_{t_2}, \ldots, u_{t_{N+1}}\}$, where $u_{t_i}$ is the system state at time $t_i$. For this paper, we assume the time lapse between consecutive states, $d_t = t_{i+1} - t_i$, are all equal and very small. Using finite difference form of Equation 4, we can predict the system state change as:

$$\Delta u'_{t_i} = d_t[\phi_1(\theta), \phi_2(\theta), \ldots, \phi_n(\theta)] \begin{bmatrix} W_1(u_{t_i}) \\ W_2(u_{t_i}) \\ \vdots \\ W_n(u_{t_i}) \end{bmatrix}.$$

On the other hand, we can obtain the true value $\Delta u_{t_i} = u_{t_{i+1}} - u_{t_i}$ of the state changes from experiment data. Hence, typically the learning algorithm minimizes the following loss function to match the predicted changes with the observed changes:

$$\min_\theta \quad L(\theta) = \sum_{i=1}^N ||\Delta u'_{t_i} - \Delta u_{t_i}||_2^2 \tag{5}$$

Stochastic gradient descent is used to minimize the aforementioned loss function.

## 4.2 RAPID-PDE: ACCELERATE LEARNING WITH RANDOM PROJECTION

RAPID-PDE makes the loss function computation and backpropagation more efficient by exploiting the sparsity of changes. It compresses the temporal updates and sparse features from a high dimensional space into a low dimensional representation using random projection. As both the expected system change and ground truth changes are sparse, the loss computation involves many redundant subtraction in the form $(0-0)$. Therefore, trivial implementation of loss function computing loss for each point in space and then summing them up introduces heavy redundant computation.

To illustrate RAPID-PDE, we assume each $u_{t_i}$ and their associated $W_1(u_{t_i})$, ..., $W_N(u_{t_i})$ are all represented as vectors. In practice, these quantities are often represented as matrices (or tensors) if the systems under consideration are 2D (or 3D). Nevertheless, it is simple to vectorize such matrices (or tensors). To avoid clutter of notations, we still use $u_{t_i}$ and $W_1(u_{t_i})$, ..., $W_N(u_{t_i})$ to represent the vector form of these matrices (tensors).

Let $\boldsymbol{P}$ be a randomly initialized projection matrix. Instead of minimizing the loss function in Equation 5, RAPID-PDE minimizes the following projected loss using SGD:

$$\min_\theta \quad L'(\theta) = \sum_{i=1}^N ||\boldsymbol{P}\Delta u'_{t_i} - \boldsymbol{P}\Delta u_{t_i}||_2^2 \tag{6}$$

**One Time Projection Overhead**. Our approach RAPID-PDE only incurs projections as a one-time pre-processing step before the beginning of training epochs. No additional projections are needed during training. Notice that $\boldsymbol{P}\Delta u_{t_i}$ is observed dynamics from data, which does not change during training. Hence $\boldsymbol{P}\Delta u_{t_i}$ can be computed in a pre-processing step. It is straightforward to verify that:

$$\boldsymbol{P}\Delta u'_{t_i} = d_t[\phi_1(\theta), \phi_2(\theta), \ldots, \phi_n(\theta)] \begin{bmatrix} \boldsymbol{P}W_1(u_{t_i}) \\ \boldsymbol{P}W_2(u_{t_i}) \\ \vdots \\ \boldsymbol{P}W_n(u_{t_i}) \end{bmatrix} \tag{7}$$

---

**Algorithm 1:** RAPID-PDE: Learning Sparse and Decomposable PDE Models from Compressed Features

---

**Input** :PDE trajectories $u_t$ at times
$\quad\quad t = 1, 2, 3, \ldots (T + 1)$, Compression ratio $r$
**Output :**PDE model parameters $\theta$

---
1: **for** $t \leftarrow 1$ **to** $T$ **do**
2: $\quad \Delta u_t \leftarrow u_{t+1} - u_t$
3: $\quad W_t \leftarrow \text{EXTRACT\_FEATURES}(u_t)$
4: **end**
5: Initialize random projection matrix $\boldsymbol{P}$ of appropriate dimension computed from original $u_t$ size and $r$
6: **for** $t \leftarrow 1$ **to** $T$ **do**
7: $\quad \Delta u_{t(compressed)} \leftarrow \boldsymbol{P} \Delta u_t$
8: $\quad W_{t(compressed)} \leftarrow \boldsymbol{P} W_t$
9: **end**
10: Initialize neural network (NN) PDE model parameters $\theta$
11: **repeat**
12: $\quad \Delta u'_{t(compressed)} = NN(W_{t(compressed)}, \theta)$
13: $\quad$ Loss $L'(\theta) =$
$\quad\quad \sum_{t=1}^{T} ||\Delta u'_{t(compressed)} - \Delta u_{t(compressed)}||_2^2$
14: $\quad$ Update $\theta$ by backpropagating error gradients $\Delta_\theta L'$
15: **until** *converge*
16: **return** $\theta$

---

Luckily, due to the separation of learning parameters with observational features, the only terms that are changing during training are $[\phi_1(\theta), \phi_2(\theta), \ldots, \phi_n(\theta)]$. Exploiting this formulation, we can pre-compute $\boldsymbol{P} W_1(u_{t_i})$, $\ldots$, $\boldsymbol{P} W_n(u_{t_i})$ in a pre-processing step as well. Only $[\phi_1(\theta), \phi_2(\theta), \ldots, \phi_n(\theta)]$ are updated during training.

Algorithm 1 shows the pseudocode of RAPID-PDE to learn Sparse and Decomposable PDEs. With a given set of PDE trajectories $u_t$ at times $t = 1, 2, \ldots, (T + 1)$, and given compression ratio of the compressed new dimensions for features, this algorithm outputs the model parameters $\theta$ that minimizes the loss function defined in Equation 6. In line $1 - 4$, we compute the changes in PDE trajectory $\Delta u_t$ and features $W_t$ for times $t = 1, 2, \ldots, T$. In line 5-9, we initialize a random matrix $\boldsymbol{P}$ of appropriate dimensions and then compress the $\Delta u_t$ and $W_t$ by random projection with $\boldsymbol{P}$. In line 10-15, we randomly initialize PDE model parameters $\theta$, then update the parameters $\theta$ by backpropagating error gradients of the loss function given in Equation 6.

## 5   THEORETICAL ANALYSIS

RAPID-PDE guarantees learning performance. We know from the Johnson-Lindenstrauss lemma Johnson and Lindenstrauss [1984] that high dimensional points can be embedded into lower dimensional space, while nearly preserving distance. For compressed sensing, restricted isometry property as introduced in Candes and Tao [2005] provides

the necessary and sufficient condition for the compressed sensing matrix, which transforms high dimensional sparse vectors into low dimensional vectors. We can construct random matrices in similar manner for our purposes as well and we will show this in this section. Such analysis are available for many other applications which also use dimensionality reduction Clarkson and Woodruff [2017], Shi et al. [2009], Razenshteyn et al. [2016]. In the following theoretic analysis, we show a constant approximation guarantee between the projected loss and the original one. Hence, an algorithm optimizing for the surrogate loss function (Equation 6) indirectly optimizes the original loss function (Equation 5). First, it is easy to see:

**Claim 5.1.** *For optimal parameter value $\theta^*$ with $L(\theta^*) = 0$, we will also have $L'(\theta^*) = 0$.*

In practice, we may not be able to find a $\theta$ which reduces the loss function to zero. However, a more careful theoretic derivation shows the projected loss function is only a multiplicative factor away from the original loss function (Theorem 5.1). Before introducing the theorem, we first introduce the notion of sub-exponential random variables, which are widely used in analyzing random projections, e.g., in Boucheron et al. [2003]:

**Definition 5.1.** *$X$ is a random variable, $E(x) = \mu$. $M_{x-\mu}(\lambda) = E[exp(\lambda(X - \mu))]$ is the moment generating function of $X - \mu$. $X$ is sub-exponential with parameter $(\sigma^2, b)$ if for all $|\lambda| < 1/b$, $\ln M_{x-\mu}(\lambda) \leq \lambda^2 \sigma^2 / 2$.*

**Theorem 5.1.** *Suppose the projection matrix $P = (p_{i,j})_{n \times d}$, $p_{i,j} = y_{i,j}/\sqrt{n}$. $y_{i,j}$ are sampled i.i.d. from a given distribution. $y_i^T = (y_{i,1}, \ldots, y_{i,d})$, $Y = (y_1, \ldots, y_n)^T$. $E(y_{i,j}) = 0$, $Var(y_{i,j}) = 1$. For any $x$, $||y_i^T x||^2 / ||x||_2^2$ is sub-exponential with parameter $(\sigma^2, b)$. All $\Delta u'_{t_i}$ and $\Delta u_{t_i}$ have at most $k$ non-zero elements, $2k < n$. $0 < \delta < \min\{1, \sigma^2/b\}$. Suppose $\theta^*$ is the optimal parameter which minimizes $L(\theta)$, i.e., $\theta^* = \arg \min L(\theta)$. Then with probability at least $1 - 2(12/\delta)^{2k} \exp(-n\delta^2/(8\sigma^2))$, we have:*

$$(1 - \delta)^2 L(\theta^*) \leq L'(\theta^*) \leq (1 + \delta)^2 L(\theta^*). \quad (8)$$

*On the opposite side, suppose $\theta'$ is the local optimal solution found by RAPID-PDE, with the same probability we have:*

$$(1 - \delta)^2 L(\theta') \leq L'(\theta') \leq (1 + \delta)^2 L(\theta'). \quad (9)$$

Theorem 5.1 guarantees that RAPID-PDE will find a solution that is at most $(1 + \delta)^2$ times the optimal solution $L(\theta^*)$, if stochastic gradient descent is not trapped in local minima when optimizing for the surrogate loss function $L'(\theta)$. Conversely, suppose RAPID-PDE finds a solution $\theta'$, if again we assume this solution is at the global minimum, then the global minimum of the original loss function $L(\theta^*)$ is at most a multiplicative factor away. The proofs of all

the theorems and corollaries are left to the supplementary materials.

We can prove more concrete guarantees based on Theorem 5.1 when $p_{i,j}$ are sampled from a few well-known probabilistic distributions. For example, when $p_{i,j}$ are sampled i.i.d. from the Gaussian distribution $N(0, 1/n)$, we have:

**Corollary 5.1.** *Suppose the projection matrix $P = (p_{i,j})_{n \times d}$, $p_{i,j}$ are sampled i.i.d. from $N(0, 1/n)$. All $\Delta u'_{t_i}$ and $\Delta u_{t_i}$ have at most $k$ non-zero elements, $k < n$, $0 < \delta < 1$. $\theta^*$ and $\theta'$ are defined the same as in Theorem 5.1, with probability $1 - 2(12/\delta)^{2k} \exp(-n\delta^2/32)$, we have:*

$$(1-\delta)^2 L(\theta^*) \leq L'(\theta^*) \leq (1+\delta)^2 L(\theta^*). \quad (10)$$

*With the same probability, we have:*

$$(1-\delta)^2 L(\theta') \leq L'(\theta') \leq (1+\delta)^2 L(\theta'). \quad (11)$$

Also, when $p_{i,j}$ are uniform i.i.d. distributed, we have:

**Corollary 5.2.** *Suppose the projection matrix $P = (p_{i,j})_{n \times d}$, $p_{i,j}$ are instead sampled i.i.d. from uniform distribution with mean $\mu$ and variance $\sigma^2$. Let $c_d = \frac{d(5d+4)}{5} - 1$, $b_d = 3d - 1$. Then, for $0 < \delta < \min\{1, c_d/b_d\}$ with probability at least $1 - 2(12/\delta)^{2k} \exp(-n\delta^2/(16c_d))$ we have:*

$$(\sigma(1-\delta) - \mu)^2 L(\theta^*) \leq L'(\theta^*) \leq (\sigma(1+\delta) + \mu)^2 L(\theta^*).$$

*With the same probability, we have:*

$$(\sigma(1-\delta) - \mu)^2 L(\theta') \leq L'(\theta') \leq (\sigma(1+\delta) + \mu)^2 L(\theta').$$

## 6 RELATED WORKS

**Learning PDEs and ODEs** PDEs arise in diverse systems and applications such as face recognition Fang et al. [2017], turbulence prediction Portwood et al. [2019], sea surface temperature forecasting de Bezenac et al. [2018] etc. Previously different approaches have been proposed to learn differential equations via machine learning. Bar-Sinai et al. [2019] proposed to replace classical fixed finite difference formulae with data driven discretization to approximate solution to PDEs. Our work is similar to Schaeffer [2017], where spatial derivatives are computed and then fitted with time derivatives using compressed sensing. Our method on the other hand assumes prior knowledge about the PDE terms, and does not need the signal recovery step via $L_1$ optimization as in compressed sensing. Neural networks, in different forms have also been explored to solve PDEs in different dynamic systems (Sirignano and Spiliopoulos [2018],Raissi et al. [2019], Lutter et al. [2018], Demeester [2019], Long et al. [2018]). Besides learning PDEs, neural networks have also been used for learning ODEs. Recent work on Neural ODEs (Chen et al. [2018]) and their variants as proposed in Kidger et al. [2020], Lee and Parish [2021], Jia and Benson [2019], Chen et al. [2020], Yin et al. [2021] aim to learn dynamics of a system using neural network. Besides neural networks, Raissi and Karniadakis [2018] assumes the PDE solution is gaussian distributed and proposes using Gaussian process for solution. Other prominent works in solving PDE systems include Han et al. [2018], Beck et al. [2019]. All the aforementioned methods use the full dimension of the data, while our algorithm first performs compression and then uses the compressed data for learning.

**Physics Learning** There have been a significant research effort in learning physics models from data such as Hamiltonian neural networks (Greydanus et al. [2019]), Lagrangian neural networks (Cranmer et al. [2020]), Deep lagrangian neural networks (Lutter et al. [2018]). Neural networks have been used to analyze in-situ experiment data for materials under extreme heat and irradiation (Niu et al. [2020]). Similar to our work, Xue et al. [2021] proposes to make physics learning more efficient by combining 2 step PDE trajectory extraction and model learning into a single learning method, and Sima and Xue [2021] proposes to use locality sensitive hashing to avoid redundant computations for similar data points make forward simulation more efficient. In our work, we assume that the PDE trajectories are already extracted, and we do not presume any knowledge about similarity of data points. Rather we only assume that the corresponding PDE model can be decomposed into sparse features and parameter functions.

**Random Projections based Methods.** Random projection has a wide variety of applications and forms the basis of techniques such as compressed sensing (Donoho [2006], Candes and Tao [2006], Candès et al. [2006]) and locality sensitive hashing (Indyk and Motwani [1998], Gionis et al. [1999]). Random projection based method is also used for classification Cannings and Samworth [2017], clustering Dasgupta [1999] and regression Dobriban and Liu [2019] as well. We only mention a few example of random projections applications here, although the total number of such works is pretty large.

## 7 EXPERIMENTS

We tested the efficacy of RAPID-PDE on learning the phase-field models of grain growth and void defect evolution for metallic materials. Overall, RAPID-PDE is able to learn models of comparable quality with algorithms without compression, even with data compressed to 0.05% of the original size. In this way, RAPID-PDE reduces training times by as much as 70% for the grain growth model and as much as 50% for the nanovoid evolution.

**Dataset.** We used synthetic data for experimenting on both grain growth and nanovoid defect evolution in ma-

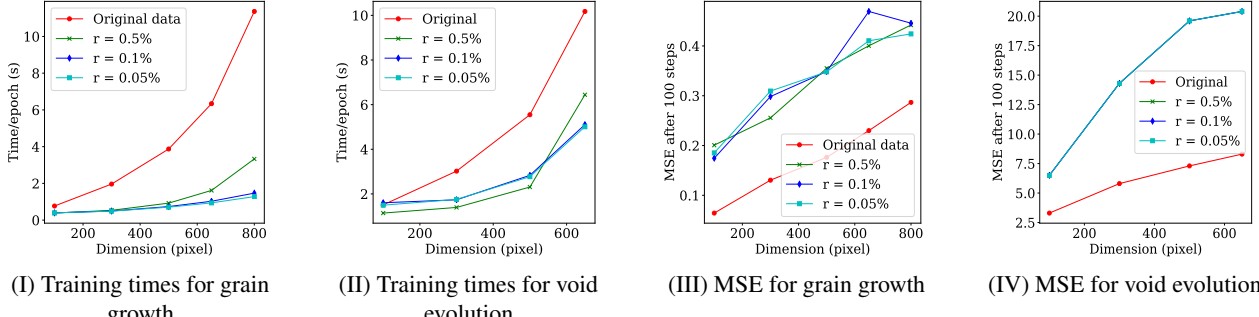

(I) Training times for grain growth    (II) Training times for void evolution    (III) MSE for grain growth    (IV) MSE for void evolution

Figure 4: RAPID-PDE algorithm saves 50% to 70% of training times compared to algorithms without compression (Figures I, II) while preserving comparable learning performance (Figures III, IV) when tested on a separate testing set. $r$ in the figures are the compression ratio to the original data. **(I-II)** Training times for grain growth and nanovoid defect evolution for 1000 epochs. **(III-IV)** Mean squared error (MSE) for grain growth and void defect evolution. The MSE was calculated by performing simulation for 100 timesteps with the same initial conditions for different algorithms and then comparing the simulated output with ground truth states. The MSE shown in **(IV)** for different $r$ values are overlapping, the learned models are very close to each other, and only provides significant difference in output when simulated for timesteps $>> 100$.

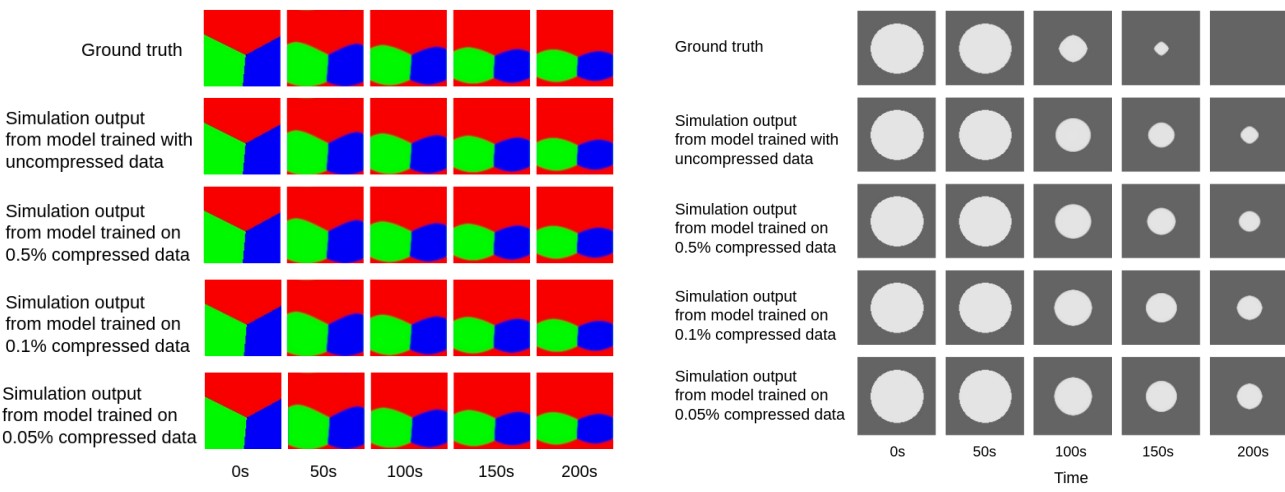

Figure 5: RAPID-PDE learns the ground truth model parameters governing the grain growth dynamics. The learned model demonstrate similar dynamics as the ground-truth model when simulating from the same initial condition.

Figure 6: RAPID-PDE learns the ground truth model parameters governing the nanovoid defect evolution. The learned model demonstrates similar dynamics as the ground-truth model when simulating from the same initial condition.

terials. For generating data for grain growth, we used the model described in Fan and Chen [1997] to simulate the dynamics of 3 grains in a specimen. For the nanovoid defect evolution, we used a simplified version of the evolution model described in Millett et al. [2011] to simulate the dynamics of a single void shaped defect. It is simplified in the sense that we excluded the random thermal fluctuation and random introduction of the void interstitial part of the model, to ensure fair comparison between our model and the baseline model. For the grain growth application, we generated data for 5 different $N \times N$ 2-dimensional grids, for $N = 100, 300, 500, 650, 800$. For

nanovoid evolution, we generated similar 2-dimensional grid data for $N = 100, 300, 500, 650$. During training we used 1000 grids/frames from simulated data, and a seperate 300 grids/frames for testing.

**Baseline method.** To ensure fair comparison, we test our method against a baseline method that does not perform any compression of the extracted features. Thus the only difference between our method and the baseline method is the presence of the intermediate compression step.

**Faster training.** Our training method using compressed data provides faster training time compared to a baseline method that uses original uncompressed data. The training times for grain growth dynamics in various dimensions

are shown in Figure 4I. For nanovoid defect evolution, the training times are shown in Figure 4II. We used uniform distribution $U(1, 50)$ to sample the projection matrix $P$. As the compressed data becomes smaller, the training times become shorter.

**High accuracy of learned model.** To evaluate the accuracy of learned models, we computed mean square error (MSE) of the simulated output with ground truth data after 100 steps of simulation. We also simulated the evolution dynamics for 200 seconds using the learned model parameters for both grain growth and void evolution. We can see the MSE for grain growth in Figure 4III and for void evolution in Figure 4IV. The simulation outputs are shown in Figure 5 and Figure 6. As we can see from these results, the MSE for both baseline method and our method are reasonably small, and the simulation results match closely with the actual ground truth data. Our method however is more efficient as it saves 50% to 70% computation time.

## 8  CONCLUSION

In this paper, we introduced RAPID-PDE, an efficient algorithm to learn Sparse and Decomposable PDEs from experimental data using random projection of features. Our method takes advantage of the sparsity of updates to compress high dimensional PDE trajectories into low dimensional representation, thus saving computation time while preserving learning performance. Experiments with two phase-field models - one for grain growth and another for nanovoid defect evolution in materials prove that our method leads to faster learning of underlying physics models, and the learned model provides reasonable matches with the ground truth observation when tested. We hope in the future, we can explore the application of RAPID-PDE to other exciting domains.

### Author Contributions

Md Nasim played the main role in conceiving the idea, writing the code and writing the paper. Yexiang Xue contributed during idea brainstorming, and theoretical analysis. Xinghang Zhang and Anter El-Azab contributed in writing, and provided the data for Figure 2II.

### Acknowledgements

This research was supported by NSF grant CCF-1918327. We thank anonymous reviewers for their comments and suggestions.

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
