# OpenReview forum: "Efficient Learning of Sparse and Decomposable PDEs using Random Projection"
_auai.org/UAI/2022/Conference — UAI 2022 Poster_

### Official Review · Reviewer_JzbK · 2022-04-10

**Q2(1) Originality/Novelty:** 3
**Q2(2) Significance/Impact:** 2
**Q2(3) Correctness/Technical Quality:** 3
**Q2(6) Clarity Of Writing:** 3
**Q6 Overall Score:** 5
**Q8 Confidence In Your Score:** 3

**Q1 Summary And Contributions:**

This paper propose an algorithm to learn sparse and decomposable PDEs from data. To this end, the key component is random projection of features.


**Q2 Assessment Of The Paper:**

More detailed information regarding each of these aspects is given below:

**Q2(4) Quality Of Experiments (Optional):**

2: Fair: The experimental evaluation is weak: important baselines are missing, or the results do not adequately support the main claims.

**Q2(5) Reproducibility:**

4: Excellent: Key resources (e.g., proofs, code, data) are available and key details (e.g., proof sketches, experimental setup) are comprehensively described for competent researchers to confidently and easily reproduce the main results.

**Q3 Main Strengths:**

This is a timely work. The work is interesting and the paper is well written.


**Q4 Main Weakness:**

The major issue in this paper is that it is missing comparative analysis. There are many related work, but the authors did not compare to any of them, including learning PDEs methods.


**Q5 Detailed Comments To The Authors:**

The authors seek to provide “efficient learning…”, but they never compare to other methods. To be efficient, this should be motivated by a comparative analysis.

We recommend the authors to provide a comparative analysis with other methods in the literature for the problem at hand.

There are many grammatical errors, such as “some grows bigger and some shrinks”, “high interest to physicist”, “these void defects causes”, “These phase-field variables are continuous, lies in range”, “Such interfaces, by nature, only accounts for”, “All of these above expression for”, “the only terms that are changing during training is”, “our algorithm first perform compression and then use”, “the training times becomes shorter”, The learned model demonstrate”, “are reasonable small”, “The learned model demonstrate”, “seperate”, “we show the how the update”, “we get what need to prove”, “Lets assume”,…


**Q7 Justification For Your Score:**

Our overall assessment is based on all the aforementioned comments.


**Q9 Complying With Reviewing Instructions:**

1: Yes.

---

### Official Review · Reviewer_PrUQ · 2022-04-15

**Q2(1) Originality/Novelty:** 2
**Q2(2) Significance/Impact:** 2
**Q2(3) Correctness/Technical Quality:** 3
**Q2(6) Clarity Of Writing:** 3
**Q6 Overall Score:** 5
**Q8 Confidence In Your Score:** 2

**Q1 Summary And Contributions:**

This paper presents a method for efficiently learning PDEs. The model extracts a set of hand-crafted features at each location, and a linear combination of these features is used to predict the change in state. The main contribution is to project both the features and the state into a random subspace. Concentration bounds from compressed sensing show that assuming sparsity of the features and transitions, the projections are sufficient to accurately recover the dynamics.

**Q2 Assessment Of The Paper:**

More detailed information regarding each of these aspects is given below:

**Q2(4) Quality Of Experiments (Optional):**

2: Fair: The experimental evaluation is weak: important baselines are missing, or the results do not adequately support the main claims.

**Q2(5) Reproducibility:**

3: Good: Key resources (e.g., proofs, code, data) are available and key details (e.g., proofs, experimental setup) are sufficiently well-described for competent researchers to confidently reproduce the main results.

**Q3 Main Strengths:**

In terms of topic, the paper is motivated by scientific applications outside the usual suspects. The potential downstream applications seem interesting (though it's hard for me to judge). The paper is well-written, and I enjoyed reading about the models and the applications. The random projection technique seems interesting, and is well motivated in terms of the sparse structure of the PDEs. It's novel as far as I'm aware (but I know very little about either PDEs or the application areas). I haven't checked the theoretical results carefully, but they seem believable and relevant to the task.

**Q4 Main Weakness:**

The main shortcoming is that it's not clear whether the proposed technique would work on real data. The experiments use simulated noiseless datasets with extremely simple generation rules. Presumably real data would be full of noise, unmodeled dynamics, distribution shift, etc. Something more naturalistic would help a lot.

Since the main motivation is computational efficiency, it would be good to have at least one experiment in a setting where computation is a bottleneck.

There needs to be a stronger baseline. What's a standard approach to this problem?  Alternatively, the task feels like an instance of autoregressive modeling of video, so it could be worth comparing against a generic autoregressive model such as an RNN.

One aspect of the algorithm that feels like it might not generalize is that it's based on predicting temporal differences in the states. Wouldn't the temporal differencing amplify whatever noise there is in the state observations?


**Q5 Detailed Comments To The Authors:**

I found it interesting to read about the application areas, but it would help to explain how the proposed method would be used in an application. E.g., would the learned PDE form the generative model as part of a larger inference algorithm?  Is the aim to learn something from simulations of a system?  This would help the reader understand the requirements (e.g. accuracy) on the algorithm and inform how it should be evaluated.

Minor typo: "reduces training times to as much as 70%": Do you mean "reduce training times by"?

I assume the experiments are run on a CPU. What sort of hardware is likely to be used for the application?  If GPUs are common in this area, would they be suited to exploit the sparse structure used in this method?


**Q7 Justification For Your Score:**

The paper is well written and there seem to be promising ideas. The theoretical results are well motivated. The main thing I'd like to see is a more ambitious and naturalistic set of experiments.

I have essentially no background in PDE modeling, so please keep that in mind.

**Q9 Complying With Reviewing Instructions:**

1: Yes.

---

### Official Review · Reviewer_8WXz · 2022-04-16

**Q2(1) Originality/Novelty:** 3
**Q2(2) Significance/Impact:** 2
**Q2(3) Correctness/Technical Quality:** 3
**Q2(6) Clarity Of Writing:** 4
**Q6 Overall Score:** 6
**Q8 Confidence In Your Score:** 4

**Q1 Summary And Contributions:**

The paper proposes an approach for learning parameters of a partial differential equation that would fit the observed data. In its core this is a deep learning approach that maps a current time state of a system along with a parameter setting to the system change relative to time. The crux of the contribution is observation that a projection matrix over system state and change/delta allow for effective computation of the loss function, which compares predicted and actual time-deltas.

**Q2 Assessment Of The Paper:**

More detailed information regarding each of these aspects is given below:

**Q2(4) Quality Of Experiments (Optional):**

2: Fair: The experimental evaluation is weak: important baselines are missing, or the results do not adequately support the main claims.

**Q2(5) Reproducibility:**

3: Good: Key resources (e.g., proofs, code, data) are available and key details (e.g., proofs, experimental setup) are sufficiently well-described for competent researchers to confidently reproduce the main results.

**Q3 Main Strengths:**

It is an interesting application and quite different problem from what I am used at seeing while doing reviewing.

The approach is sound and the use of a projection matrix can reduce the dimension of the problem. The operating assumption is that time-change affects small number of system components and the intrinsic dimension of the problem should be low.

**Q4 Main Weakness:**

Theoretical analysis could be related to a number of ML approaches for dimensionality reduction. For instance, I could see similarities with hashing kernels and low-rank approximation of inner products. The analysis does not appear to have a starting point nor strong reference to prior work and yet it does not appear groundbreaking novel.

Empirical results are not very convincing for problems when the dimension of instance space is large. In fact, I would not expect such a strong divergence in Figure 4-iv) for cases where changes are sparse and intrinsic dimension is quite low.

The illustrated simulations appear to be quite simple and the discussion could be more elaborate on why they were selected (e.g., relative to theoretical results). I would expect some more complex looking simulation as an evidence that the approach works.

**Q5 Detailed Comments To The Authors:**

I find that the paper is well written and organised. The problem is interesting and I would say it was not easy to present this to ML community (PDEs are not that abundant). The illustrations of the neural architecture and the way deep learning fits the bill are also good and satisfactory.

The related work is a bit thin on approaches relying on projection matrices and randomised algorithms as a mean of finding an approximate solution.

**Q7 Justification For Your Score:**

The paper is different from anything in my batch and for that "novelty" reason think it might be interesting to have it presented at the conference. I feel it might better connect people working on inverse problems in physics and ML.

**Q9 Complying With Reviewing Instructions:**

1: Yes.

---

### Official Review · Reviewer_z1Ue · 2022-04-18

**Q2(1) Originality/Novelty:** 2
**Q2(2) Significance/Impact:** 2
**Q2(3) Correctness/Technical Quality:** 1
**Q2(6) Clarity Of Writing:** 3
**Q6 Overall Score:** 4
**Q8 Confidence In Your Score:** 4

**Q1 Summary And Contributions:**

The paper discuss how random projections can be used to improve the process of learning sparse and decomposable PDEs (a class of partial differential equations). To this end, the paper notices that the trajectories of these certain PDEs can be compressed with random projections into much lower dimension without a substantial change to the learning objective. The paper gives a theoretical analysis as well as an overview of empirical validation.

**Q2 Assessment Of The Paper:**

More detailed information regarding each of these aspects is given below:

**Q2(4) Quality Of Experiments (Optional):**

4: Excellent: The experimental evaluation is comprehensive and the results are compelling.

**Q2(5) Reproducibility:**

4: Excellent: Key resources (e.g., proofs, code, data) are available and key details (e.g., proof sketches, experimental setup) are comprehensively described for competent researchers to confidently and easily reproduce the main results.

**Q3 Main Strengths:**

- empirical verification of the presented claims
- good writing quality and illustrations
- reproducibility: the code attached
- possible impact on some applied researchers (learning PDEs)

**Q4 Main Weakness:**

My concern is that the theoretical contribution is rather weak:
- the central claim would be a straightforward application of the Restricted Isometry Property; furthermore I believe the current analysis doesn't deliver what is promised and extra work on the proof has to be done.
- auxiliary facts and definitions are missing the credits/citations to the theory literature; in fact I don't find even a credit to Johnson-Lindenstrauss :-(

**Q5 Detailed Comments To The Authors:**

Introduction - the summary of the main result claims that the logarithmic number of features is sufficient; from the formula it follows that n should be rather poly-logarithmic (as the term n delta^2 needs to be much bigger than log(1/delta))

Definition 5.1 - the notion should be credited to the relevant literature. Here you discuss *sub-gamma distributions* (sub-exponential is also used albeit less common in probability literature), see the book "Concentration Inequalities" by S. Boucheron, G. Lugosi and P. Massart.

Random Projections on Sparse Vectors - consider crediting and utilising explicitly the notions of "k-sparse vectors" and "Restricted Isometry Property".

Theorem 9.1 -  This kind of bound is very popular and I think the broader audience would benefit from some explanation. The "union-bound term" before the exponential term appears due to covering: by scaling you can consider only the sphere, and on the sphere with k fixed non-zero coordinates delta/2-covering is of size O(1/delta)^k.

Corollary 5.1 and Corollary 5.2 - please consider crediting these facts to the literature as these parameters have been studied already; also please give credits to the properties such as "Chernoff Bounds for sub-gamma/sub-exponential random variables".

Theorem 5.1 - I believe the current proof doesn't deliver what is promised. First, we are missing the dependency on time (union bound?) as the probabilistic property is invoked at every time step. Second, the proof assumes a limited sparsity definition namely that the non-zero locations of vectors are fixed (different than in the related literature!). A more careful argument should be carried out to allow non-zero positions to be anywhere and to address these, possibly inspired by other technical facts on the Restricted Isometry Property.


**Q7 Justification For Your Score:**

I like the idea of giving random projections one more application. However in my opinion the quality of the presented theoretical analysis is not sufficient.

**Q9 Complying With Reviewing Instructions:**

1: Yes.

---

### Decision · Program_Chairs · 2022-05-15

**Decision:**

Accept (Poster)

**Comment:**

Meta Review: The focus of the submission is the learning of physics models automatically from data, more specifically sparse and decomposable partial differential equations (PDEs, (4)). The authors propose to solve a random projected variant of the PDE [(6)] and show constant approximation guarantees under mild assumptions (Theorem 5.1). The efficiency of the approach is illustrated in the context of grain growth models and nanovoid defect evolution.

Data-driven learning of PDEs has various successful applications in machine learning and physics. As it was assessed by the reviewers the authors present a valuable contribution at the interface of these areas. They also pointed to the fact that the work could have been strengthened somewhat by a more thorough comparison with related work in the ML literature (both at conceptual and numerical level).